# Development of a Prototype Observatory of Heat-Related Occupational Illnesses and Injuries through the Collection of Information from the Italian Press, as Part of the WORKLIMATE Project

**DOI:** 10.3390/ijerph20054530

**Published:** 2023-03-03

**Authors:** Giulia Ionita, Michela Bonafede, Filippo Ariani, Alessandro Marinaccio, Marco Morabito, Miriam Levi

**Affiliations:** 1Medical Specialization School of Hygiene and Preventive Medicine, University of Florence, 50134 Florence, Italy; 2Occupational and Environmental Medicine, Epidemiology and Hygiene Department, Italian Workers’ Compensation Authority (INAIL), 00143 Rome, Italy; 3CeRIMP (Regional Centre for Occupational Injuries and Disease of Tuscany), Local Health Authority Tuscany Centre, 50135 Florence, Italy; 4Institute of Bioeconomy, National Research Council (IBE-CNR), 50019 Florence, Italy; 5Epidemiology Unit, Department of Prevention, Local Health Authority Tuscany Centre, 50135 Florence, Italy

**Keywords:** heat-related illness, heat stress, news, occupational injuries, press, workers’ health, workplace

## Abstract

Exposure to heat is a recognized occupational risk factor. Deaths and accidents at work caused by high temperatures are underestimated. With the aim of detecting and monitoring heat-related illnesses and injuries, a prototype database of occupational events attributable to critical thermal conditions reported in Italian newspapers was created. Information was analyzed from national and local online newspapers using a web application. The analysis was conducted from May to September during the three-year period 2020–2022. Articles concerning 35 occupational heat-related illnesses and injuries were selected; 57.1% of the events were reported in 2022, and 31.4% of total accidents occurred in the month of July 2022, when the Universal Thermal Climate Index daily mean values corresponded to “moderate heat stress” (51.0%) and “strong heat stress” (49.0%). Fatal heat-related illnesses were the most frequent conditions described. In most cases, workers had been involved in outdoor activities in the construction sector. A comprehensive report was created by compiling all relevant newspaper articles to enhance awareness of this issue among relevant stakeholders and promote heat-risk prevention strategies in the current context where heatwaves are becoming increasingly frequent, intense and long-lasting.

## 1. Introduction

Climate change is the primary cause of the increased frequency of extreme weather conditions such as heatwaves, floods and wildfires [1]. Events such as heatwaves play an important role in population health, and studies confirm a general increase in heat-related mortality [2,3,4,5]. Indeed, a reduced capacity to respond and adapt to extreme heat increases the risk of organ damage due to exceeding physiological thermoregulatory capacity [6]. Exposure to excessive heat is also a well-known occupational hazard. Workers under heat stress conditions appear to be four times more likely to experience heat strain compared to individuals working in an environment with neutral temperatures [7]. Heat-related occupational health risks are exacerbated during activities carried out in sunny outdoor environments and in indoor workplaces when lack of ventilation, a poor cooling system or processes that generate heat do not allow proper regulation of temperatures [8]. Workers in agricultural and construction sectors are among those most exposed, with clear evidence also at the Italian level [9,10,11], especially those with jobs requiring high levels of physical exertion, the use of personal protective equipment and/or heavy clothes that prevent heat loss [12]. Negative health effects arising on account of dehydration and overheating, such as sweating, dizziness, poor sleep quality, and physical and mental exhaustion, with impairment of reasoning and increased reaction times, may increase the risk of injury [13]. Moreover, short-term heat-related illnesses, such as heat cramps, heat exhaustion and heat stroke may also arise [14]. If not adequately and promptly treated, heat stroke is a fatal condition. Some workers have a greater susceptibility to heat-related illness; factors such as pre-existing heart and respiratory diseases, taking certain medications (hypotensive drugs, diuretics, sedatives, etc.) [15], or being pregnant or disabled increase the risk associated with exposure to high temperatures [16].

Heat exposure makes workers a vulnerable population; depending on the work situation, they often lack the authority to control their exposure time, find a shaded area [17], arrange for refreshing breaks, and maintain constant access to water. Additionally, in cases of undeclared work, they are often not covered by employment injury insurance. Such occupational conditions make certain categories of workers more subject to the negative effects of heat exposure in the workplace and contribute to heat-related productivity losses [18]. Occupational exposure to extreme temperatures exacerbates social inequalities, especially in relation to conditions such as low income, ethnic status, low literacy and education. Workers employed in precarious, strenuous, risky, low-tech, and unskilled jobs, such as migrant workers, may suffer greater job insecurity, which is associated with increased heat-related morbidity and mortality [16,19].

As part of the WORKLIMATE Project (“Impact of environmental thermal stress on workers’ health and productivity: intervention strategies and development of an integrated heat and epidemiological warning system for various occupational sectors”) [20], a study was conducted during the warmer months of 2020–2022 to assess the effectiveness of using online newspapers as a tool for promptly detecting and monitoring heat-related illnesses and injuries in the workplace in Italy, as well as for fostering interventions to protect the health of workers exposed to heat.

## 2. Materials and Methods

News articles regarding the effects of extreme thermal conditions on workers’ health published in the Italian daily press during the warm months (May to September) in the three-year period 2020–2022 were quantitatively and qualitatively assessed. The search for articles was set up using a web application accessible via web browser (VALIRIA, developed by the Joint Laboratory of Technological Solutions for Clinical Pharmacology, Pharmacovigilance and Bioinformatics of the University of Florence), that allows the configuration and execution of customized queries to be launched on the Google search engine. The search strategy performed was the following: [*(“climate change” OR “killer heat” OR “scorching heat” OR temperature* OR “global warming” OR sultriness OR hot OR drought) AND (work OR worker* OR “construction site” OR “day laborer” OR farmer OR company OR tractor OR farming OR garden) AND (“heat stroke” OR accident OR injury OR “sudden illness” OR dead OR die OR fall)]*. A daily report of search results was automatically sent by email to two researchers (G.I. and M.L.). The search was conducted in both national and local online newspapers, chosen because of their large readership as assessed by ADS, the association that publishes data on the circulation of the daily and periodical press in Italy [21]. According to October 2022 data from Audipress, the official reference survey of readership of the daily and periodical press in Italy, 21.8% of Italians, corresponding to more than 11 million readers, read one of the main newspapers on paper or digital every day. The steady growth in the reading of digital editions was also confirmed, with an estimated audience of 6.5 million readers [22]. Articles were included if they focused on heat-related illnesses and injuries among workers due directly to high temperature exposure. In addition, a manual search was conducted every two weeks to verify that all relevant news had been captured by the web application. Articles with no mention of extreme hot conditions or work environment were excluded.

The events were classified as “injuries” in the case of traumatic events, or as heat-related illnesses, based on the description provided in the article. For each article, whenever available, information about sex, age, nationality, Italian region in which the accident occurred, labour sector in which the worker was employed, activity performed immediately before the event and severity (fatal versus non-fatal) of heat-related illnesses/injuries was collected. The occupational sector of each worker was classified according to the ATECO classification of economic activities adopted by the Italian National Institute of Statistics-Istat [23]. Descriptive statistical analyses were performed using frequency measures and contingency tables.

Data on air temperature (°C), relative humidity (%), wind speed (m s^−1^) and solar global radiation (W m^−2^) on the relevant days were collected from meteorological stations located near the municipality where the heat-related events (illnesses or injuries) were reported by the Italian daily press. Hourly meteorological data were gathered from “Weather Underground” [24] and were used to calculate a thermal comfort index, the Universal Thermal Climate Index (UTCI) (by using the UTCI software code “version a 0.002”, freely available online, http://www.utci.org/), currently considered the state of the art for outdoor biometeorological indices [25]. UTCI is an equivalent temperature (°C) referring to a person generating 135 W m^−2^ (therefore a value on the border between a low and moderate metabolic rate) based on the most recent scientific progress in human thermophysiology, biophysics, and heat exchange theory [26]. UTCI categorizes thermal stress in ten classes (from extreme cold to extreme heat stress conditions) relating to heat stress, starting from a situation of “no thermal stress” (9 °C < UTCI ≤ 26 °C), through “moderate heat stress” (26 °C < UTCI ≤ 32 °C), “strong heat stress” (32 °C < UTCI ≤ 38 °C), “very strong heat stress” (38 °C < UTCI ≤ 46 °C) and “extreme heat stress” (UTCI > 46 °C). For the days in which heat-related events occurred, the daily average and maximum UTCI values and the respective categories are shown.

## 3. Results

On average, the web application mail report showed five to thirty daily articles, with peaks detected during heatwaves. The final report containing selected articles was published on the WORKLIMATE project homepage (https://www.worklimate.it/ (accessed on 7 February 2023)) for free access [27,28].

Cases that appeared in newspapers from May 2020 to September 2022 were selected. If reported in multiple articles, the event was counted only once. All links to articles in which the incident was mentioned were reported in the final reports. According to web-based press and inclusion criteria, 35 workers suffered from health outcomes related to occupational heat stress in Italy in the three-year period considered. The days on which the selected events occurred were characterized in 51.0% of cases by daily mean UTCI values corresponding to “moderate heat stress” (daily mean UTCI of 29.6 °C ± 1.5 °C) and in the remaining 49.0% on days with daily mean UTCI values corresponding to “strong heat stress” (daily mean UTCI of 33.7 °C ± 1.1 °C). Furthermore, the maximum daily UTCI values were predominantly (80.0% of days) values corresponding to the “very strong heat stress” class (the average daily maximum UTCI value was 42.7 °C ± 1.9 °C) and on 14.0% of days the daily maximum UTCI values were in the “extreme heat stress” class (the average daily maximum UTCI value was 48.7 °C ± 0.9 °C) (Table 1).

Heat-related illnesses were the most reported (*n* = 32; 91.4% of all reported events), with only three being injuries (Table 2). There were 19 fatal events, corresponding to 54.3% of all heat-related events. Among the fatal events, all except one were caused by heat-related illnesses, the remaining death being ascribed to a fall. Both the daily mean (32.1 °C) and maximum (44.3 °C) UTCI values on days with fatal events were on average higher by 1.4 °C and 2.5 °C respectively than the mean and maximum UTCI values on days with non-fatal events (30.8 °C and 41.8 °C for mean and maximum daily UTCI values, respectively). Furthermore, all the events that occurred on days when the maximum daily value of UTCI reached “extreme” heat stress levels (UTCI > 46 °C) resulted in fatalities.

In 26 cases (74.3% of all reported cases) the activities were carried out outdoors and in nine cases the laborers were working in indoor environments, mainly sheds. For one event, neither the activity nor the occupational sector in which the victim was involved were indicated.

In terms of the distribution of events over time, four heat-related illnesses/injuries were reported in 2020, corresponding to 11.4% of the events that occurred in the three-year period considered (Figure 1). In 2021, 11 cases were reported (31.4% of the events that occurred in the three-year period 2020–2022); of these, almost all (*n* = 9; 81.8%) occurred during a heatwave in June. Finally, 20 events (57.1% of the events that occurred in the three-year period) were reported in 2022, particularly during heatwaves in July, when 11 events were reported (55.0% of events occurred in 2022 and 31.4% of events occurred in 2020–2022).

Only one event was reported in May 2022 and no events were reported in the month of September during the three-year period.

Of the 35 workers involved, 31 (88.6%) were men and only 4 (11.4%) were women. Regarding nationality, 12 (34.3%) were Italian, 8 (22.9%) were of foreign nationality and for 15 workers (42.9%) their nationality was not mentioned in the news.

Almost half (40.0%) of the events involved middle-aged workers (30–59 years) (Table 3). Only two workers (5.7%), both of them foreigners, were younger than 30. No worker was older than 70. The ages of 12 men and 3 women (corresponding to 42.9% of all workers) were unknown.

Events occurred throughout Italy, except for five regions (Aosta Valley, Abruzzo, Molise, Basilicata, Sardinia), where no events were reported in the online newspapers. Apulia recorded the highest number of heat-related illnesses/injuries in the examined period (*n* = 8; 22.9%) (Figure 2).

The highest number of events was reported in the construction sector (*n* = 11; 31.4% of all recorded events), followed by agriculture and forestry (*n* = 8; 22.9% of all recorded events) and manufacturing activities (14.3%). Traumatic events (injuries) were recorded in construction activities and for an airport service worker (included in “other service activities”) (Figure 3).

## 4. Discussion

Environmental temperatures and the number of heatwaves have been increasing each year, and the three years in which we conducted the study (2020–2022) were among the warmest on record, according to the World Meteorological Organization [29]. Although exposure to high temperatures is considered an occupational risk factor and preventive measures for reducing the hazard of developing heat-related pathologies are often easy to implement, 35 events were reported by the Italian press in the studied period. As demonstrated by this study, all 35 events occurred on critical thermal stress days, characterized by daily average UTCI values classified as “moderate” or “strong” heat stress conditions and maximum daily UTCI values prevalently characterized by “very strong” or “extreme” heat stress conditions. Furthermore, all events that occurred on days with “extreme” heat stress conditions resulted in fatalities. Most of the workers involved were males employed in the construction sector and performing their tasks in an outdoor environment, which is in agreement with findings from the scientific literature [30,31,32]. However, no information is provided regarding workers’ prior work experience, individual training level [33,34] or degree of heat acclimatization, despite studies reporting that most fatalities occur during the first week of work in the heat, when the body has not yet adapted to high temperatures [15,35]. Our findings related to occupational sectors are consistent with the existing literature, which indicates that outdoor male workers face a higher risk of injuries during heatwaves [36]. Although it is well known in the scientific literature that young, less experienced workers are particularly susceptible to heat-related occupational injuries [10], only two workers were younger than 30 among the reported cases. As with cases involving women, the proportion of events in this age group was minimal, since older age groups and men are still more represented in at-risk jobs. In addition to the cases reported in Table 1, several additional events indirectly linked to global warming occurred as well. In August 2021, two deaths occurred in the agricultural sector. These involved a 30-year-old man who was crushed by a tractor while putting out a fire in Sicily and a 42-year-old man swept away by a landslide while draining water after a flood in the Trentino-Alto Adige region. In June 2022, in Piedmont (Northern Italy), a 57-year-old man experienced a heat stroke while burning brushwood that resulted in an organ failure requiring him to undergo a liver transplant. In the province of Florence in Tuscany (Central Italy), several healthcare workers fell sick in the operating room of a hospital due to a cooling system malfunction. In July 2022, due to abnormal heat, an avalanche occurred on the Marmolada massif in the Alps (Trentino-Alto Adige), killing three mountain guides.

The number of heat-related occupational illnesses and injuries reported in the news appreciably increased from 2020 to 2022, so much so that the number of reported events in July 2022 alone was equivalent to that of the entire previous year.

This trend testifies to the progressive recovery of commercial and industrial activities in Italy after the lockdowns caused by the COVID-19 pandemic, as well as to the fact that the summer of 2022 was the hottest in history in Europe, according to the Copernicus Global Climate Highlights 2022 report [37]. In Italy, however, the summer of 2022 was the second hottest ever (second only to 2003), with an average temperature of just over 2 °C higher than the period 1991–2020, as highlighted by the Institute of Atmospheric Sciences and Climate of the National Research Council. One event occurred in May 2022, the hottest since 2003, with record drought across many Italian regions [38].

During the three-year observational period, only four news articles described an event as a “heat stroke”. Furthermore, for the events that occurred in the summer of 2022, which was characterized by prolonged and intense heatwaves, not a single article used this term. Instead, the majority of the articles used the term “*malore*”, which translates to “sudden illness”. With the aim of communicating news more effectively to laypeople, journalists use terms that are easily understood, even if they are not entirely appropriate [39]. Many articles among those emailed daily were disregarded, as they lacked explicit mention by the reporters of the association between the events and heat, despite circumstances such as the time of the day implying a possible link. Even during heatwaves, some journalists resorted to the phrase “for reasons yet to be ascertained”, perhaps due to infrequent interviews with the workers involved or their colleagues [33]. This dearth of information could lead readers to underestimate the risks associated with heat exposure in the workplace.

Unfortunately, it is not possible to accurately identify heat-related occupational illnesses or injuries in Italian administrative healthcare databases, such as hospital discharge or emergency records, as they generally only report the ICD code of the health condition affecting the patient, with no indication of the environmental factors that may be involved [40].

By examining the INAIL database for the period of 2010–2021 and taking into account the causes of injuries or acute illnesses resulting directly from heat exposure, such as heat strokes, sunstrokes, and other effects of extreme temperatures, a total of 569 cases were identified. Among these cases, 25 were fatal, accounting for 0.014% of all occupational injuries and 0.35% of fatalities during the same period [41].

Those direct effects do not capture all heat-related injuries. In particular, there may be indirect links, due, for example, to the reduction of attention caused by the heat.

For this reason, it is necessary to study the general correlation between ambient temperature and overall injury rates to correctly consider all the numerous ways in which heat can impact the health of workers. In a study assessing the correlation between ambient temperature and occupational injuries in Italy, considering injuries occurring from 2006 to 2010, extreme heat temperature exposure resulted in a 0.14% attributable fraction of work-related injuries for outdoor workers [10].

A recent systematic review confirmed that the risk of occupational injuries increases by 1% for every 1 °C rise in environmental temperature and by 17.4% during heatwaves, defined as several consecutive days with temperatures above the average for the period [42].

The present study has some limitations. It was not possible to monitor all newspapers, especially local ones and those requiring a subscription to access. Also, only newspaper coverage data were considered, excluding other mass media that provide information to the public, such as radio programs.

Nonetheless, we have showed that the monitoring of newspapers represents a valuable tool for tracking heat illnesses, which were given less attention in recently published estimates that mainly focused on traumatic events [9,10].

The monitoring of newspapers has also been useful for raising awareness about heat-related illnesses in the workplace among workers themselves and relevant stakeholders, thanks to the strong and extensive information capacity of the media, with the ultimate goal of prompting the implementation of heat stress prevention measures in occupational settings. This was the case in Italy where, in 2021, a young agricultural worker’s demise was widely reported, drawing the attention of stakeholders. In response, the Governor of Apulia passed an ordinance prohibiting fieldwork between 12.30 pm and 4.00 pm until the end of August, leveraging the heat stress forecasting system developed in WORKLIMATE [30] for identifying potential risks among outdoor laborers. Subsequently, agricultural activities were banned on days identified as “high risk” based on the project’s findings. The measure was adopted by three other southern Italian regions, namely Basilicata, Calabria, and Molise, in the same year, and the ordinances were renewed in 2022.

Given the impact of heat stress on health [43], the development of prevention measures is fundamental. Awareness, correct perception and knowledge of the risks related to exposure to high temperatures is necessary to be able to effectively implement prevention and management measures in the workplace [44]. Implementing preventive measures can minimize health risks associated with exposure to heat. These measures are supported by clinical evidence [45] and can be enhanced through the use of technology to facilitate the development of recommendations and guidelines. Key strategies for preventing heat-related health problems in workers include [43,45]: limiting direct exposure to heat, scheduling breaks, consuming adequate amounts of water regularly, using appropriate personal protective equipment compatible with the thermal environment, identifying workers vulnerable to heat stress who may benefit from temporary work restrictions, and promoting mutual supervision among workers [15]. In Italy, operational guidance was issued in summer 2021 for the prevention of occupational risk due to physical factors, such as microclimates, to enhance the effectiveness of preventive measures [46]. The WORKLIMATE project also developed operational strategies, such as a forecasting alert platform based on the Wet Bulb Globe Temperature (WBGT) parameter, which provides regional and sub-regional heat stress predictions for up to three days and recommendations to mitigate health effects for workers engaged in outdoor work [47]. In addition, informational brochures were developed [48] to educate workers on how to deal with occupational risk conditions. These interventions have been proven to effectively reduce the risk of heat-related illness. In fact, a randomized trial found that the group that received an “intervention package” with behavioral preventive measures experienced a 63% reduction in the risk of heat stress compared to the group not receiving the intervention [49].

## 5. Conclusions

In summary, we have examined how the media portrays the workplace, including its associated risks and problems. Media outlets may emphasize certain aspects over others, which can influence public perception and understanding of an issue [33]. How the media frames critical public health concerns, such as injuries and illnesses related to heat stress among workers, is crucial in informing the public about hazards [50]. Wakefield et al. argue that the reporting of an event “diagnoses, evaluates, and prescribes solutions to social problems” [51]. In their study, the authors view newspaper reports as a means to quickly draw attention to heat-related injuries and fatalities in the workplace among official bodies, the public, and workers themselves, as official statistics on occupational illnesses and injuries are not updated frequently enough. Furthermore, newspaper articles can promote preventative measures, which can lead to positive changes in minimizing high-risk behaviors [52].

The aim of the working group is to extend the monitoring of heat-related injury news coverage to social media platforms, such as Twitter or Facebook, in the near future.

## Figures and Tables

**Figure 1 ijerph-20-04530-f001:**
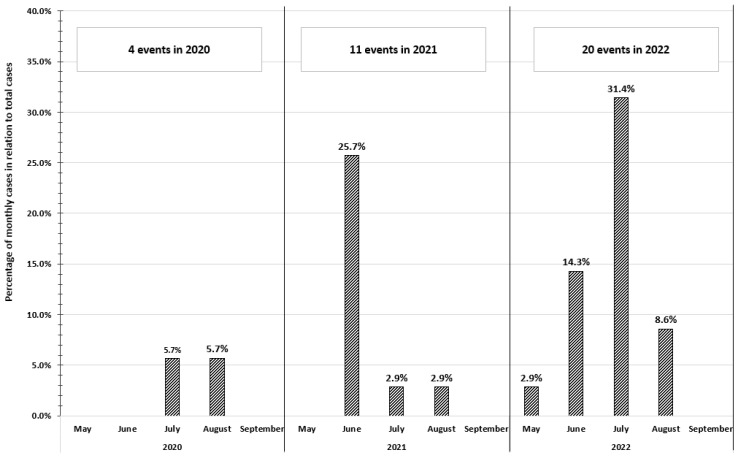
Distribution of events reported by Italian online newspapers in the years 2020 to 2022, by month and year of publication.

**Figure 2 ijerph-20-04530-f002:**
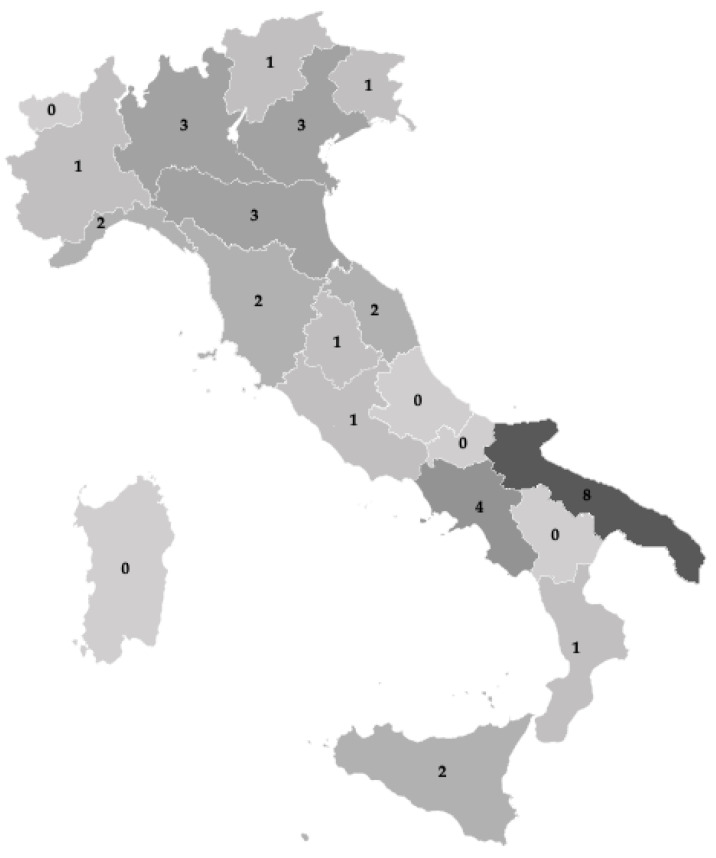
Number of occupational heat-related illnesses and injuries as reported by Italian online newspapers in the years 2020 to 2022, by region.

**Figure 3 ijerph-20-04530-f003:**
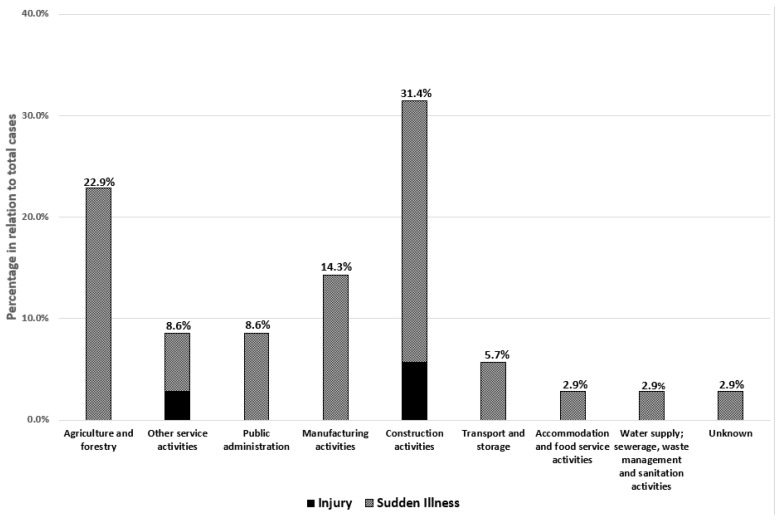
Number of occupational heat-related illnesses and injuries as reported by Italian online newspapers in the years 2020 to 2022, by occupational sector.

**Table 1 ijerph-20-04530-t001:** Summary table of heat-related illnesses and injuries in the workplace reported in the Italian national press. Details of published data—years 2020–2022. (All website accessed on 7 February 2023).

	Year, Month	Age (Years)	Sex	Nationa- lity	Occupational Sector, Activity	Region (Municipa- lity)	Event (Illness or injury) and Severity	Link to Online Newspaper	Daily Average Heat Stress Category and UTCI (°C)	Daily Maximum Heat Stress Category and UTCI (°C)
1	2020, July	55	Male	Italian	Agriculture and forestry, gardener	Lazio (Latina)	Illness, fatal	http://bit.ly/3ERtz05	Moderate 29.4	Very strong 41.5
2	2020, July	53	Male	Polish	Construction activities, worker engaged in canal reclamation	Emilia- Romagna (Bologna)	Illness, fatal	http://bit.ly/3B2RWqG	Strong 35.2	Extreme 49.6
3	2020, July	36	Male	Romanian	Construction activities, fiber optic placement	Friuli- Venezia Giulia (Pasiano)	Illness, fatal	http://bit.ly/3B1epUO	Moderate 27.9	Very strong 40.7
4	2020, August	-	Female	Italian	Public administration, municipality employee	Tuscany (Viareggio)	Illness, non-fatal	http://bit.ly/3ulSiEQ	Strong 32.4	Very strong 41.5
5	2021, June	-	Male	-	Construction activities, worker on construction site	Apulia (Taranto)	Illness, non-fatal	http://bit.ly/3B0Vs4P	Strong 32.3	Very strong 42.6
6	2021, June	-	Male	-	Construction activities, worker on construction site	Apulia (Taranto)	Illness, non-fatal	http://bit.ly/3B0Vs4P	Strong 32.9	Very strong 44.3
7	2021, June	-	Male	-	Construction activities, worker on construction site	Apulia (Taranto)	Illness, non-fatal	http://bit.ly/3B0Vs4P	Strong 32.9	Very strong 44.0
8	2021, June	-	Male	-	Construction activities, worker on construction site	Apulia (Taranto)	Illness with onset of coma, non-fatal	http://bit.ly/3B0Vs4P	Strong 33.5	Very strong 44.0
9	2021, June	38	Male	Italian	Transport and storage, tanker truck driver	Apulia (Brindisi)	Illness, fatal	http://bit.ly/3UDMNMR	Moderate 31.9	Extreme 47.4
10	2021, June	27	Male	Mali	Agriculture and forestry, day labourer	Brindisi (Apulia)	Illness, fatal	http://bit.ly/3VFtmUC	Strong 34.6	Extreme 48.3
11	2021, June	35	Male	Italian	Other service activities, leafleting	Apulia (Galatina)	Illness, fatal	http://bit.ly/3it03pY	Moderate 30.6	Very strong 42.1
12	2021, June	-	-	-	Agriculture and forestry, harvesting of agricultural products	Veneto (Province of Verona)	Illness, fatal	http://bit.ly/3VIqpCN	Moderate 29.6	Very strong 41.5
13	2021, June	-	-	-	Agriculture and forestry, harvesting of agricultural products	Veneto (Province of Verona)	Illness, non-fatal	http://bit.ly/3VIqpCN	Moderate 29.8	Very strong 41.7
14	2021, July	42	Male	Italian	Construction activities, working on a scaffold	Sicily (Palermo)	Injury (Fall), fatal	http://bit.ly/3ueoU3B	Moderate 31.6	Very strong 42.2
15	2021, August	62	Male	Italian	Agriculture and forestry, forestry worker	Apulia (Bitonto)	Illness, fatal	http://bit.ly/3ugMWLb	Strong 36.1	Extreme 49.6
16	2022, May	-	Male	-	Construction activities, working on a scaffold	Umbria (Terni)	Illness, non-fatal	http://bit.ly/3W5Zh0Y	Moderate 26.1	Very strong 43.4
17	2022, June	47	Female	Italian	Water supply; sewerage, waste management and sanitation activities, ecological worker	Tuscany (Prato)	Illness, fatal	http://bit.ly/3Vr97KK	Moderate 31.2	Very strong 46.1
18	2022, June	65	Male	Italian	Construction activities	Lombardy (Jerago con Orago)	Injury (Fall), non-fatal	http://bit.ly/3hx5mVo	Moderate 30.1	Very strong 43.0
19	2022, June	-	Female	Italian	Public administration, judge	Lombardy (Bergamo)	Illness, non-fatal	http://bit.ly/3v2wgY5	Strong 32.2	Very strong 45.3
20	2022, June	49	Male	Italian	-	Campania (Casagiove)	Illness, fatal	http://bit.ly/3HCP33X	Strong 33.3	Very strong 44.2
21	2022, June	45	Male	-	Manufacturing activities	Emilia- Romagna (Castelvetro di Modena)	Illness, non-fatal	http://bit.ly/3VHTldZ	Moderate 28.2	Very strong 39.2
22	2022, July	59	Male	Italian	Agriculture and forestry, day labourer	Calabria (Rossano)	Illness, fatal	http://bit.ly/3uTATUz	Strong 34.1	Very strong 43.9
23	2022, July	-	Male	-	Public administration, municipal employee	Campania (Battipaglia)	Illness, non-fatal	http://bit.ly/3WFUZxx	Moderate 29.5	Strong 37.0
24	2022, July	-	Female	French	Other service activities, model for fashion shows	Sicily (Siracusa)	Illness, non-fatal	http://bit.ly/3VoR481	Moderate 30.4	Very strong 40.8
25	2022, July	20	Male	Albanian	Agriculture and forestry, day labourer in a greenhouse	Campania (Falciano del Massico)	Illness, fatal	https://bit.ly/3XSDknp	Moderate 27.9	Very strong 40.5
26	2022, July	54	Male	Romanian	Construction activities, electrician working on a roof	Liguria (La Spezia)	Illness, fatal	http://bit.ly/3GZVNsn	Strong 33.6	Very strong 43.2
27	2022, July	61	Male	Italian	Manufacturing activities	Piedmont (Rivoli)	Illness followed by head injury, fatal	http://bit.ly/3BI5R5T	Moderate 31.0	Very strong 42.6
28	2022, July	-	Male	-	Manufacturing activities	Trentino—Alto Adige (Arco)	Illness followed by head injury, fatal	http://bit.ly/3Up1fI8	Strong 35.0	Very strong 43.8
29	2022, July	47	Male	Moroccan	Accommodation and food service activities, dishwasher	Liguria (Diano Marina)	Illness, fatal	http://bit.ly/3V3g83e	Strong 33.5	Very strong 44.6
30	2022, July	-	Male	-	Transport and storage, bicycle rider	Lombardy (Milan)	Illness, non-fatal	http://bit.ly/3FibcTp	Strong 33.2	Very strong 45.5
31	2022, July	67	Male	-	Construction activities, worker on a roof	Emilia- Romagna (San Donnino)	Illness, fatal	http://bit.ly/3gLhvFW	Strong 32.9	Very strong 44.3
32	2022, July	-	Male	African origin	Agriculture and forestry, day labourer in a greenhouse	Campania (Parete)	Illness, fatal	http://bit.ly/3XSDknp	Strong 34.7	Extreme 48.4
33	2022, August	50	Male	-	Manufacturing activities, worker in a shed	The Marche (Ancona)	Illness, non-fatal	http://bit.ly/3uX2Hre	Moderate 29.2	Strong 38.0
34	2022, August	30	Male	-	Manufacturing activities, shipyard worker	The Marche (Ancona)	Illness, non-fatal	http://bit.ly/3uX2Hre	Moderate 29.6	Very strong 38.3
35	2022, August	-	Male	-	Other service activities, airport baggage loading/unloading attendant	Veneto (Venezia)	Injury (ankle fracture), non-fatal	http://bit.ly/3gROBUp	Moderate 28.9	Very strong 40.9

UTCI: Universal Thermal Climate Index.

**Table 2 ijerph-20-04530-t002:** Severity of heat-related illnesses and injuries as reported by Italian online newspapers in the years 2020 to 2022.

	Fatal		Non-Fatal		Total	
	N	%	N	%	N	%
Heat-related Illnesses	18	51.4%	14	40.0%	32	91.4%
Injuries	1	2.9%	2	5.7%	3	8.6%
Total	19	54.3%	16	45.7%	35	100.0%

**Table 3 ijerph-20-04530-t003:** Demographic characteristics (gender, age and nationality) as reported by Italian online newspapers in the years 2020 to 2022, by age group.

Age Groups	Nationality						Gender				Total	
	Italian		Foreigner		Unknown		Male		Female			
	N	%	N	%	N	%	N	%	N	%	N	%
<30 years	0	0.0%	2	5.7%	0	0.0%	2	5.7%	0	0.0%	2	5.7%
30–39 years	2	5.7%	1	2.9%	1	2.9%	4	11.4%	0	0.0%	4	11.4%
40–49 years	3	8.6%	1	2.9%	1	2.9%	4	11.4%	1	2.9%	5	14.3%
50–59 years	2	5.7%	2	5.7%	1	2.9%	5	14.3%	0	0.0%	5	14.3%
60–69 years	3	8.6%	0	0.0%	1	2.9%	4	11.4%	0	0.0%	4	11.4%
Unknown	2	5.7%	2	5.7%	11	31.4%	12	34.3%	3	8.6%	15	42.9%
Total	12	34.3%	8	22.9%	15	42.9%	31	88.6%	4	11.4%	35	100.0%

## Data Availability

Not applicable.

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
