# Peer review of "Development of a Prototype Observatory of Heat-Related Occupational Illnesses and Injuries through the Collection of Information from the Italian Press, as Part of the WORKLIMATE Project"

_ijerph, 2023, doi:10.3390/ijerph20054530_

Round 1

Reviewer 1 Report

Attention is needed to terminology.  Terms such as “illness”, “injury”, and “accidents” need to be applied consistently and correctly.  For example in line 131 the term accidents is used where the figure of 11 appears to refer to the total number of illnesses and injuries combined. In line 228 the term “cases” is used: does this refer to injuries only?

It is important to state the criteria for defining an injury as heat-related.  As the authors point out (lines 234-237) the relationship between heat stress and occupational injury is calculated probabilistically by plotting injury rates over time against environmental heat parameters. In this analysis the authors appear to have found a total of only 3 cases of heat-related injury. This implies that in these cases the authors were able to conclude that but for the level of heat stress the injury would not have occurred.  What criteria were applied to make this conclusion in these three cases? Comparing the finding of 3 heat-related injuries in three years with the studies cited as references 10, 34 and 35, it appears that reliance on newspaper stories results in a significant under-enumeration of heat-related occupational injuries.

Details of the distribution of illness and injuries by region and nationality of the victim are of limited interest to readers outside Italy. On the other hand  there are no data on the variables that matter most: the environmental parameters such as air temperature, radiant temperature, humidity and wind speed.  The authors state that correlation with meteorological data was not possible since the exact date, time and location of the events could not be confirmed.  It is difficult to understand how these simple details could not be obtained by simple inquiry.  Without this information it is not possible to be confident that an illness was heat-related.

Reviewer 2 Report

In this manuscript, the authors use online newspapers to assess the occurrence of heat-related morbidity and mortality among workers in Italy. The findings reveal that newspapers and potentially other media sources may be a positive way to track and report on heat-related morbidity and mortality. Based on this they conclude that news media can influence public opinion on occupational health and safety issues when it comes to hazards like extreme heat.

I am encouraged by this manuscript because it does provide additional evidence on how climate events like extreme heat will have an impact on public health, in particular occupational health. This will influence how we may be able to perform work under current and future climate conditions given the increasing threat of climate change and associated hazards. The authors provide a novel way of tracking heat-related morbidity and mortality in occupational settings through analysis of the news and look forward to when they extend this monitoring to social media platforms like Twitter and Facebook which can provide a wealth of data.

That said, there are also parts of the manuscript that could be improved, and provide a number of specific comments below:

-In the introduction, around lines 44-68, the authors do well in describing the potential risk factors among workers, but fail to describe more specific factors around race/ethnicity, immigration status, wages/income, and type of work (e.g. heavy manual labor) which are significant contributors to heat-related illnesses among workers. It seems the authors have collected some of this data from around the world so I would point them to the 2021 US Environmental Protection Agency document ‘Climate Change and Social Vulnerability in the United States’ which has a section on extreme heat and occupations. Expanding this to include more widely held views on vulnerability among this population would be beneficial.

-The authors identify the sources as ‘national and local online newspapers’ but do not specifically signify what number or percentage of Italians these sources are reaching. Can the authors provide quantitative estimates of the reach or readership of these sources? It would be helpful in determining if this is a representative sample of new sources (via online newspapers).

-In at least two instances in the manuscript the authors refer to ‘hot’ temperatures (lines 136-138 or say ‘just over X’ temperature measure (lines 207-210), suggesting these temperatures were higher than baseline. It would be beneficial for the authors to use average temperatures during those time periods and signify how ‘hot’ or how much over baseline these temperatures were to orient the reader to their significance.

-I have concerns over the broad generalization made in lines 223-223 where it is stated that it is not possible to identify heat-related occupational illnesses or injuries in Italian administrative healthcare databases. The literature cited is from a database that ended in 2003 and is now two decades old. Citation #33 uses hospitalization data itself but makes a broad claim about the lack of data due to the rareness of heat-related illness hospitalizations. This may have been the case prior to the mainstream discussions and research around climate change and extreme heat but it may be that a significant time has passed which could warrant further investigation into these claims. Given the international audience of this journal, it may warrant further discussion on why this is the case. Is it an inability to capture this data or that there have been no significant epidemiological studies on hospitalization data, or…?

-Lines 230-234 appear unclear to the reader. The quoted statistic of 0.14%, attributed to workers exposed to extreme temperatures outdoors, is worded as if it is compared to something, but it is unclear, particularly when you identify that it is ‘confirming the relevance of indirect effects’. Please clarify.

-The authors allude to the importance of the media in potentially distributing knowledge about extreme heat impacts to working populations but there is no clear context of how this is done, who this may reach, and what effect it may have on occupational heat-related injury & illness. Is this just a screening method in lieu of hospitalization studies? What effect should the media have on this topic, particularly on the general public? How can the media be utilized to this knowledge for the benefit on worker health and safety, particularly when it comes to extreme heat? It would help to clarify the significance of these media analyses and public health actions/policies that can leverage this analysis.

Round 2

Reviewer 1 Report

This paper has been substantially revised and my major concerns have been addressed.  As suggested in my initial review, the newspaper reports need to be followed up by  obtaining the meteorological records for the relevant time, data and location.  The authors have now shown that this is feasible.  The relevant heat stress parameters are now provided.

Occupational heat-related injuries are probably dealt with separately from illnesses.  Whereas occupational illness can usually be identified by clinical means (eg hyperthermia), injuries look the same whether  they are heat-related or not, so it is not possible to be sure that the injury would not have occurred but for the influence of heat stress.  Nevertheless this is not a significant issue in this paper since there are only 3 cases purporting to be heat-related injuries.

Reviewer 2 Report

Thank you for addressing the reviewer comments and making the necessary improvements to the paper.